# Comment on Martínez-Delgado et al. Using Absorption Models for Insulin and Carbohydrates and Deep Leaning to Improve Glucose Level Predictions. *Sensors* 2021, *21*, 5273

**DOI:** 10.3390/s24134361

**Published:** 2024-07-05

**Authors:** Josiah Z. R. Misplon, Varun Saini, Brianna P. Sloves, Sarah H. Meerts, David R. Musicant

**Affiliations:** 1Department of Computer Science, Carleton College, Northfield, MN 55057, USA; misplonj@carleton.edu (J.Z.R.M.); sainiv@carleton.edu (V.S.); slovesb@carleton.edu (B.P.S.); 2Population Health, Epic Systems, Verona, WI 53593, USA; 3Neuroscience Program and Department of Psychology, Carleton College, Northfield, MN 55057, USA; smeerts@carleton.edu

**Keywords:** deep learning, LSTM, data processing, glucose level prediction, insulin absorption, carbohydrate absorption, machine learning models, model validation, time-series data

## Abstract

The paper “Using Absorption Models for Insulin and Carbohydrates and Deep Leaning to Improve Glucose Level Predictions” (*Sensors* **2021**, *21*, 5273) proposes a novel approach to predicting blood glucose levels for people with type 1 diabetes mellitus (T1DM). By building exponential models from raw carbohydrate and insulin data to simulate the absorption in the body, the authors reported a reduction in their model’s root-mean-square error (RMSE) from 15.5 mg/dL (raw) to 9.2 mg/dL (exponential) when predicting blood glucose levels one hour into the future. In this comment, we demonstrate that the experimental techniques used in that paper are flawed, which invalidates its results and conclusions. Specifically, after reviewing the authors’ code, we found that the model validation scheme was malformed, namely, the training and test data from the same time intervals were mixed. This means that the reported RMSE numbers in the referenced paper did not accurately measure the predictive capabilities of the approaches that were presented. We repaired the measurement technique by appropriately isolating the training and test data, and we discovered that their models actually performed dramatically worse than was reported in the paper. In fact, the models presented in the that paper do not appear to perform any better than a naive model that predicts future glucose levels to be the same as the current ones.

## 1. Introduction

Type 1 diabetes mellitus (T1DM) is an autoimmune disease that leads to the destruction of the insulin-producing beta cells of the pancreas, leaving people with T1DM unable to regulate their blood glucose levels without exogenous insulin injections [1]. Maintaining euglycemia, which is a concentration of blood glucose between 70 and 180 mg/dL [2], is a daily challenge for people with T1DM. Both hypoglycemia and hyperglycemia have negative health consequences and can lead to severe complications, including death [3,4,5]. Advances in technology, such as continuous glucose monitoring systems (CGMs), rapid-acting insulin, and insulin pumps, have dramatically improved the time in range and reduced diabetes-related complications [6]. In particular, sensor-augmented pumps that employ algorithms to suspend or increase insulin when low or high blood glucose is predicted have reduced the frequency and severity of hypo- and hyperglycemic events [7,8].

The article “Using Absorption Models for Insulin and Carbohydrates and Deep Leaning to Improve Glucose Level Predictions” by Martínez-Delgado et al. [9] addresses the need for better algorithms by using exponential-shaped curves to simulate carbohydrate and insulin absorption in the body. The paper explains that training a long short-term memory neural network (LSTM) on raw carbohydrate and insulin data yields a root-mean-square error (RMSE) on predicted glucose values of 15.5 mg/dL, whereas training on data that are fitted to exponential curves yields an RMSE of 9.2 mg/dL. Our team was impressed with these results, and thus, we reached out to Martínez-Delgado et al. in hopes of applying their algorithm to our research questions, and they generously provided us with their code repository. We discovered that Martínez-Delgado et al.’s model validation scheme was flawed; it mixed training data and test data chosen from the same time intervals, which is problematic when doing autoregressive forecasting on time-series data [10]. After we corrected this measurement error in their code, we discovered that their models actually performed dramatically worse than reported in the original paper. In fact, the predictive models presented in the Martínez-Delgado et al. paper do not appear to perform any better than a naive model that predicts future glucose levels to be the same as current ones. The purpose of this comment is to document that the models presented in the Martínez-Delgado et al. paper were not, in fact, shown to make reliable predictions of glucose levels.

## 2. Experimental Design, Both Original and Corrected

In order to develop a machine learning model, a dataset is conventionally broken into three parts: a training set, a validation set, and a test set. The primary reason for separating the data into distinct subsets is to avoid overfitting the model, in favor of allowing it to learn broader trends for application outside of the specific dataset. The training set is used to teach the model how to interpret data and predict results based on that data. The validation set is used to evaluate how effective a model is and further tune it to make better predictions. After the model has completely finished training, it is applied to the test set to evaluate its accuracy.

Martínez-Delgado et al. randomly assigned data points to three different sets, using 60% of their data for the training set and 20% each for the validation and test sets. In most applications, the ordering of the data does not matter in any way, and thus, random selection is a simple and appropriate approach for breaking the data into subsets [10]. However, Martínez-Delgado et al. missed a critical subtle issue. Martínez-Delgado et al.’s experiment used time-series data from the D1NAMO dataset [11], which Martínez-Delgado et al. [9] transformed into sliding windows that each contained two hours of information (such as blood glucose levels and insulin values). The dataset they created consisted of many data items that were not independent from each other. In fact, many distinct entries in the data that they used were extremely highly correlated with each other by design. This led to issues when two overlapping windows were separated into distinct sets—specifically, one into the training set and one into the test set—because the model evaluated itself on data that were nearly identical to the data on which it was trained. This is a problem that was observed in other instances concerning time-series data [10]. As a result, the performance on the test data was not representative of the performance on unseen data.

Here is some more detail to help clarify the above. Each instance of data used for training a given model represented a two-hour window, with model-specific features being reported at 5-min intervals. Depending on the model, features could include raw or processed information about calorie intake, insulin administered, accelerometer data, and blood glucose measurements. (Accelerometer data was included in all of the models.) One instance of data, therefore, might have started at 9:00 and ranged through to 10:55. A second instance that started at 9:05 and ranged through to 11:00 would then overlap heavily with the first. Random selection for the test and training sets could result in the 9:00–10:55 instance being used in the training set and the 9:05–11:00 instance being used in the test set. This means that in the Martínez-Delgado et al. approach [9], the instance being tested was nearly identical to the instance in the training set, and thus, using it as a test point incorrectly tested the model for true predictions of the future.

To prevent this bleeding between test and training data, we used a different approach to split the data into training, validation, and test sets: we allocated the first 60% of the data to the training set, the following 20% to the validation set, and the final 20% to the test set. By preserving the temporal nature of the data, we ensured that there was no overlap between the training and test sets, as the validation set functionally served as a buffer between them. This update that we made corrected the errors in the measurement techniques in the original study, and thereby demonstrated that the true RMSE values for the models were substantially worse than those reported by Martínez-Delgado et al. Moreover, our correction showed that the experimental comparison presented by Martínez-Delgado et al. was invalid. We summarize these results below.

## 3. Results

For clarity, we report our results in units of mmol/L (rather than mg/dL), which is consistent with how the majority of values are expressed in the original paper.

We first executed the code provided to us by Martínez-Delgado et al. to attempt to duplicate the results presented in the paper before making any of our recommended changes. We did need to make some inconsequential changes to the code, such as setting new file system paths. Our fork of the code with these minor changes is visible on GitHub (https://github.com/carleton-diabetes-data-sci/PrediccionGlucosa, accessed 19 April 2024). Since the provided code randomly initializes the neural networks and randomly shuffles the data without setting a global seed, it is essentially impossible to duplicate the precise numbers shown in the paper.

Table 1 shows the results of our rerunning of the code across the seven different approaches presented by Martínez-Delgado et al. [9] (referred to in their paper as “tries”). Likely due to the randomization issues mentioned above, our numbers were not precisely the same as those in their paper. As we will see, however, these differences were minimal when compared with the effects of correcting the randomization procedure. For clarity, we refer to the results presented by Martínez-Delgado et al. [9] as “original” and the results from our rerunning of the code as “replicated”.

We then corrected the evaluation code so that the training, validation, and test sets were no longer temporally intermixed and otherwise retained the methods of Martínez-Delgado et al. [9] in full. Upon running the code with this methodology (referred to here as “corrected”), we observed that the RMSE values were actually dramatically worse than the incorrect values that were reported in the original paper. The replicated and correct RMSE values can be seen in Table 2.

It is important to note the scale of the differences in the RMSEs in what Martínez-Delgado et al. originally reported when compared with the corrected RMSE values. According to the American Diabetes Association, the target blood sugar range for most people is between 3.9 mmol/L (70 mg/dL) and 10 mmol/L (180 mg/dL) (https://diabetes.org/about-diabetes/devices-technology/cgm-time-in-range, accessed 19 April 2024). In other words, an interval whose width is approximately 6.1 mmol/L (approximately 110 mg/dL) spans the entire range of normal blood glucose values. In this context, it is evident that the scale of the error that Martínez-Delgado et al. [9] made was considerable. The inaccurate results presented by Martínez-Delgado et al. [9] indicate RMSE values across the seven “tries” of approximately 0.4–0.6 mmol/L (7.2–10.8 mg/dL). An error in prediction of 0.6 mmol/L (10.8 mg/dL) would be considered relatively small in the context of the range that is considered to be normal. However, when we corrected the experiments, we discovered that the actual RMSE values across the seven “tries” ranged from approximately 3.8 to 4.9 mmol/L (approximately 68–88 mg/dL). This level of error is dramatically bigger. An error in prediction of 4.9 mmol/L (88 mg/dL) is approaching the magnitude of the entire range of normal glucose levels.

In addition to the scale of the RMSE values changing notably when we repaired the experimental technique, we also observed a change in the conclusion regarding which modeling approach was best. In the original paper [9], Martínez-Delgado et al. claim that try 5 performed the best when averaged across all the patients. While this was visible in our replicated numbers, this appeared to be an inaccurate conclusion because no such pattern emerged in our runs with corrected data splitting. In fact, try 5 emerged as the second-worst approach.

The corrected RMSE values that we measured on the models from the Martínez-Delgado et al. study [9] had a dramatically higher magnitude than the original paper RMSE values, which raised concerns as to whether the models in the original paper succeeded at all. To evaluate this, we applied a naive approach to predicting blood glucose levels. Given the past two hours of blood glucose data, our naive approach simply predicted that the blood glucose level one hour from now will be exactly the same as the most recent value in the two-hour window, ignoring all other data (insulin, calorie intake, etc.). In other words, this model predicted that the blood glucose level at any given time will be identical to what it was one hour earlier. This naive approach to prediction resulted in an average RMSE of 2.40 mmol/L across patients. This was dramatically lower than any of the corrected RMSE values in Table 2, indicating that the more complex neural network models presented by Martínez-Delgado et al. [9] were in fact less effective than an extremely simplistic algorithm. One could ask the following: why do the approaches presented in the Martínez-Delgado et al. paper [9] generalize more poorly than a simple naive approach? A full answer to that question is beyond the scope of this work, but a reasonable hypothesis would be that the Martínez-Delgado et al. models heavily overfit the training data, and that this critical error was missed because of the incorrect splitting of the training and test data.

In summary, we demonstrated that the accuracy levels of the modeling approaches proposed by Martínez-Delgado et al. [9] were measured incorrectly in their study. As a result, the conclusions reached in that paper regarding the relative performances of their seven different modeling approaches were invalid because the results radically changed when the experimental procedure was corrected. Moreover, it appeared that the approaches in that study may perform no better than a rudimentary naive approach when applied to the D1NAMO dataset. Therefore, our work additionally serves as a useful demonstration of the importance of publishing open-source code and datasets in that it enables replication and scrutiny of published work.

## Figures and Tables

**Table 1 sensors-24-04361-t001:** The mean RMSEs (mmol/L) of the original results from Martínez-Delgado et al. [9] and our replicated results.

	Try 1	Try 2	Try 3	Try 4	Try 5	Try 6	Try 7
Original	0.86	0.58	0.58	0.59	0.51	0.57	0.52
Replicated	0.86	0.61	0.57	0.58	0.49	0.55	0.51

**Table 2 sensors-24-04361-t002:** The mean RMSEs (mmol/L) of our replicated results and our corrected methodology’s results.

	Try 1	Try 2	Try 3	Try 4	Try 5	Try 6	Try 7
Replicated	0.86	0.61	0.57	0.58	0.49	0.55	0.51
Corrected	3.76	4.17	4.27	4.35	4.64	4.92	4.39

## Data Availability

Information about the D1NAMO dataset can be found in Reference [11], and our GitHub repository is located at the following URL: https://github.com/carleton-diabetes-data-sci/PrediccionGlucosa (accessed 19 April 2024).

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
