# Peer review of "Comment on Martínez-Delgado et al. Using Absorption Models for Insulin and Carbohydrates and Deep Leaning to Improve Glucose Level Predictions. Sensors 2021, 21, 5273"

_sensors, 2024, doi:10.3390/s24134361_

Round 1

Reviewer 1 Report

Comments and Suggestions for Authors

This commentary by Misplon and co-workers is a critique and re-analysis of a recent paper that presented a deep learning approach for predicting blood glucose levels in subjects with type 1 diabetes with seemingly excellent predictive power. The commentary convincingly demonstrates that the reported predictive power is an artefact resulting from a flawed method evaluation based on highly correlated training and test sets.

The paper is very well written and provides an excellent explanation of the consequences of incorrectly splitting correlated data such as time-series into training and test data. The authors also report a re-evaluation of the method using a correct data split, which results in abysmal predictive power that is even outperformed by a simple last-value-carried-forward approach. 

With more and more user-friendly ML toolboxes becoming available and current increased interest in machine learning methods in diabetes research, this paper is a timely reminder and an excellent case study of the many pitfalls that await when evaluating ML and statistical models. The presented case of highly correlated time-series data is particularly relevant as these data are the workhorse for training predictive models in diabetes research. Implicitly, the paper also demonstrates the importance of publishing open source code alongside publications as well as the importance of openly accessible datasets for making research fully reproducible (and allowing detailed scrutiny). 

One point the author might want to consider in their discussion is the fact that the data is also clustered by subject, which would enable distinguishing predictive power within- and between-subjects, i.e., comparing a population-based from a personalized approach.

Reviewer 2 Report

Comments and Suggestions for Authors

I don't see any value in this comment. The authors replicated the work of a published paper by applying a slight modification in the training-validation-testing split. The author affirm that they found that a simple naive approach can perform better than the approaches presented in the original paper. However, I don't see any statistical evidence that support this statement, but just a simple number, which is irrelevant. I would recommend the rejection of this work.

Reviewer 3 Report

Comments and Suggestions for Authors

The claim made by the authors of the present paper is that Martínez-Delgado et al.'s model validation scheme, which involved mixing training data and test data chosen from the same time intervals, is problematic when performing autoregressive forecasting on time series data. The authors believe that this methodology is not suitable for the type of data they are working with.

The methodology proposed by the authors of the present paper, which involves splitting the data into training, validation, and test sets by allocating the first 60% of the data to the training set, the following 20% to the validation set, and the final 20% to the test set, aims to address the problem of temporal overlap in the data splitting scheme used by Martínez-Delgado et al. By preserving the temporal nature of the data, the authors intend to eliminate the issue of the model evaluating itself on data that is nearly identical to the data on which it was trained.

The methodology proposed by the authors can be expected to reduce the problem of temporal overlap and make the test data more representative of unseen data compared to the random selection method used by Martínez-Delgado et al. However, the authors themselves acknowledge that their proposed correction substantially worsens RMSE values compared to those reported by Martínez-Delgado et al. This suggests that their correction results in a significant degradation of the model's performance.

While the proposed methodology may indeed address the problem of temporal overlap and may reduce the problem of temporal overlap, it does so at the cost of significantly degrading the model's performance and appears to introduce new challenges or limitations that affect the model's predictive accuracy negatively.

The authors of the present paper do not claim that their method improves the model's performance; instead, they highlight that their method invalidates the experimental comparison presented by Martínez-Delgado et al. This suggests that finding an appropriate data splitting strategy for time series data, especially in the context of autoregressive forecasting, remains a challenging and important issue. Also it is essential to consider that research findings can vary, and further studies may be needed to confirm these results.

Round 2

Reviewer 2 Report

Comments and Suggestions for Authors

The authors addressed my concerns. I am in favour of pubblication.

Reviewer 3 Report

Comments and Suggestions for Authors

1. Line 73: Is there a requirement for accelerometer data?

2. Line 93: If the data is randomly shuffles, how does your previous claim that training and test data set overlap hold true?

3. There are no curves to show that the model was overfitting during validation (because if there is an issue with the dataset, even validation dataset should have the same issue and the plotted curves must show the overfitting behaviour of the model.

4. The value of RMSE can range from 0 to positive infinity. The claim that the values have a dramatically higher magnitude does not seem very convincing.

Round 3

Reviewer 3 Report

Comments and Suggestions for Authors

The authors have satisfactorily addressed all queries.